# Comparison of Intensive Care Scoring Systems in Predicting Overall Mortality of Sepsis

**DOI:** 10.3390/diagnostics15131660

**Published:** 2025-06-29

**Authors:** Mustafa Ozgur Cirik, Guler Eraslan Doganay, Melek Doganci, Tarkan Ozdemir, Murat Yildiz, Abdullah Kahraman, Seray Hazer, Mehtap Tunc, Kerem Ensarioglu, Azra Ozanbarci, Oral Mentes

**Affiliations:** 1Department of Anesthesiology and Reanimation, Ankara Ataturk Sanatorium Training and Research Hospital, University of Health Sciences, Sanatoryum St., Kecioren 06290, Ankara, Turkey; gulerdoganay@hotmail.com.tr (G.E.D.); melekdidik@hotmail.com (M.D.); mehtaptunc@hotmail.com (M.T.); 2Department of Pulmonology, Ankara Ataturk Sanatorium Training and Research Hospital, University of Health Sciences, Sanatoryum St., Kecioren 06290, Ankara, Turkey; tabiptarkan@hotmail.com (T.O.); drmuratyildiz85@gmail.com (M.Y.); kerem.ensarioglu@gmail.com (K.E.); 3Anesthesiology and Reanimation Intensive Care Unit, Ministry of Health Ankara Etlik City Hospital, Kecioren 06170, Ankara, Turkey; abdullahhero100@gmail.com (A.K.); azraarslan@yahoo.de (A.O.); 4Department of Thorasic Surgery, Ankara Ataturk Sanatorium Training and Research Hospital, University of Health Sciences, Sanatoryum St., Kecioren 06290, Ankara, Turkey; drserayhazer@gmail.com; 5Intensive Care Unit, Gulhane Training and Research Hospital, University of Health Sciences, Kecioren 06010, Ankara, Turkey; omentes@live.com

**Keywords:** APACHE II, intensive care scoring systems, mortality, OASIS, SAPS-II, SOFA

## Abstract

**Background:** Prognostic scoring systems are applied in intensive care units (ICUs) to monitor patients’ responses to treatment and guide treatment modalities. These scoring systems are also used as predictors in sepsis, where mortality rates are high. This study aims to compare the scores (APACHE II, SOFA, SAPS II, OASIS) in terms of their role in predicting overall mortality in patients admitted to ICUs with a diagnosis of sepsis or septic shock. **Methods:** Among 740 patients admitted to the tertiary intensive care unit within a 2-year period, 165 patients diagnosed with sepsis and septic shock were included in the study. Demographic data, comorbidities, SOFA, SAPSII, OASIS, and APACHE II scores, invasive or noninvasive mechanical ventilation requirements and durations, ICU admissions, hospital stays, and 28-day mortalities were retrospectively evaluated. **Results:** All scoring systems were positively correlated with mortality and CCI score. SAPS II and OASIS showed a higher correlation with mortality compared to other scoring systems, correlated with ICU admission and mechanical ventilation, unlike other scoring systems. The AUC values for the APACHE II, SOFA, SAPS II, and OASIS were 0.803, 0.873, 0.902, and 0.879, respectively. No statistically significant difference was found between the scores (*p* > 0.05). **Conclusions:** Compared to commonly used scoring systems, OASIS is a practical tool and serves as a robust scoring system for assessing mortality in ICU patients diagnosed with sepsis.

## 1. Introduction

Intensive care unit (ICU) severity scores are crucial tools for evaluating the severity of illness in critically ill patients, informing clinical decision-making, and predicting patient outcomes. These scoring systems evaluate various physiological parameters to estimate the risk of mortality and the degree of organ dysfunction. As the average age of patients in intensive care units increases, the need for ideal, highly prognostic systematic models for assessing intensive care severity has also increased. However, these models often include variables that are difficult to obtain within the first 24 h after admission [1,2,3]. Complex scoring systems are challenging to implement in clinical practice because they require the scoring of multiple measurements.

It is important to note that no ICU severity scoring system is perfect, and each has its limitations. Additionally, several factors affect mortality, including lead-time bias, pre-ICU location, acute diagnosis, physiological reserve, and patients’ preferences for life support. Although it is unlikely that the systems used will include all predictor variables, the high correlation between model simplicity and performance reflects the desired scoring model [4,5,6,7,8].

The most widely used and studied Intensive Care severity scores are the Acute Physiology and Chronic Health Evaluation (APACHE) II and the Sequential Organ Failure Assessment (SOFA) score. However, the Simplified Acute Physiology Score (SAPS) II and the Oxford Acute Illness Severity Score (OASIS) are also widely used.

The Acute Physiology and Chronic Health Evaluation (APACHE) II consisted of 12 physiological measurements, age, previous health status, and the ICU admission diagnosis. The 12 physiologic variables were heart rate, mean arterial blood pressure, temperature, respiratory rate, alveolar to arterial oxygen tension gradient, hematocrit, white blood cell count, creatinine, sodium, potassium, pH/bicarbonate, and Glasgow Coma Scale (GCS) score [7]. The SOFA scoring scheme assigns 1 to 4 points to each of the following six organ systems on a daily basis depending on the level of dysfunction: respiratory, circulatory, renal, hematology, hepatic, and central nervous system [8]. The SAPS II (Simplified Acute Physiology Score) uses similar data to APACHE II, collecting the worst physiological and biochemical variables within the first 24 h, as well as evidence of chronic disease and age. It also incorporates the type of ICU admission (planned/unplanned and medical or surgical). The OASIS does not require laboratory tests or imaging examinations, making it a practical option for rapid assessment. It evaluates parameters such as age, Glasgow Coma Scale score, mean arterial pressure, and reason for ICU admission to predict patient outcomes.

Predicting a condition’s prognosis is an important topic, especially in cases of those with high mortality, such as sepsis. In intensive care units (ICU), prognostic scoring systems have been implemented to observe patients’ treatment responses and guide medical practitioners on treatment modalities [9]. Different scoring systems are used to assess the severity of disease in intensive care patients. Determining intensive care severity scores is very useful in clinical practice [10].

A former scoring system, APACHE II, was defined in 1985 by Knaus et al. with a similar goal in mind, albeit with a different approach [11]. The score, proposed as an admission evaluation to predict overall risk in critical care, is designed to be calculated within the first 24 h of ICU admission. With a range of 0 to 71, the more severe the patient is deemed, the higher the overall score. Currently, this scoring system is among the most commonly used in ICUs. However, APACHE II is more complex than other scoring systems due to the abundant parameters it requires for assessment.

In 1993, Le Gall, Lemeshow and Saulnier proposed a scoring system to evaluate overall mortality and morbidity in a patient [12]. This scoring system, known as the Simplified Acute Physiology Score (SAPS II), involves a comprehensive assessment of physiological changes in a patient to evaluate their status, including demographic information, hospitalization etiology, comorbidities, and cardiac or respiratory support requirements. The Simplified Acute Physiology Score (SAPS II) provides an estimate of the risk of mortality without requiring a primary diagnosis and is regarded as a starting point for evaluating the efficiency of ICU [13].

Vincent et al. defined the Sequential Organ Failure Assessment (SOFA) in 1996 to evaluate patients in critical condition due to sepsis, which includes vital signs, clinical observations, and laboratory parameters [10]. The scoring system assigns a score between 0 and 4 to six categories, including cardiovascular, respiratory, and central nervous systems, with additional parameters for coagulation status and liver and renal function test results. An increase of two or more points in SOFA is attributed to the presence of sepsis in a patient.

Johnson et al. proposed another scoring system in 2013 that does not require laboratory parameters used in intensive care and includes fewer parameters and is based on clinician observations for ease of use [14]. This scoring system aimed to provide similar accuracy results with complex models by working with fewer variables. This study showed that OASIS has similar predictive power to more complex systems such as APACHE II, which require less data. The OASIS can be calculated with 10 parameters such as age, heart rate, respiratory rate, Glasgow coma score without the need for laboratory tests. With this feature, it stands out as a practical tool, especially in cases where resources are limited or rapid assessment is required. Developed in the United Kingdom to assess disease severity in critically ill patients in intensive care units (ICUs), OASIS can be a practical option for rapid assessment since it does not require laboratory tests or imaging examinations. This study showed that the OASIS has similar predictive power to more complex systems such as APACHE II, with less data requirements [14]. Another study has shown that OASIS can effectively predict mortality in a variety of patient populations, including patients with sepsis and acute stroke [15].

Complex scoring systems are challenging to implement in clinical practice because they require the scoring of multiple measurements. OASIS is a scoring system that does not involve laboratory tests or imaging examinations. OASIS is widely used for the differential diagnosis of acute disease severity and has also been confirmed to have high identification and calibration efficiency for the prognosis of ICU patients. OASIS may replace the more complex existing prediction systems [16].

For instance, a study published in Frontiers in Neurology in 2023 highlighted the predictive value of OASIS in acute stroke patients with stroke-associated pneumonia [17]. OASIS included ten components, seven of which were physiological parameters, and the remaining three were age, elective surgery, and duration of former hospitalization [18,19]. An optimal scoring system is needed to create accurate prognostic predictions for patients.

This study aims to compare the discussed scores in their role of predicting overall 28-day mortality in patients admitted to ICU with a diagnosis of sepsis or septic shock.

## 2. Materials and Methods

Patients admitted to our center’s tertiary intensive care from the emergency department or inpatient service unit between 1 April 2018 and 30 April 2020 were included in the study. A total of 740 patients were admitted during the specified period, and their computer and file records were reviewed.

For sepsis definition, quick SOFA parameters were utilized, which included presence of hypotension (systolic BP below or equal to 100 mmHg), impaired consciousness (Glasgow Coma Score below 13), tachypnea (above 22 per minute), and patients who had at least two of these parameters were categorized under sepsis classification. The definition of septic shock requires the presence of sepsis, with additional requirements for vasopressors (to keep a mean arterial pressure above 65 mmHg) and a lactate level of above 2 mmol/L. Sepsis diagnosis was established based on Sepsis-3 criteria, considering clinical judgment, microbiological culture results, imaging studies, and laboratory findings. In cases where microbiological confirmation was not possible, the diagnosis was made based on strong clinical suspicion and multidisciplinary consensus among attending intensivists and infectious disease specialists.

A total of 112 patients with a score of 2 or higher in q-SOFA were included in the sepsis category and accepted for the study. An additional 53 patients under the septic shock definition were included in the study for a total of 165 patients.

Exclusion criteria were defined and included patients without sepsis at the time of admission, those in the ICU for less than 24 h, those with inadequate parameters for proper evaluation, and those younger than 18 years.

Demographic data, comorbidities, results of SOFA, SAPS-2, OASIS, and APACHE II scores, invasive or noninvasive mechanical ventilation requirements and durations if required, ICU admission durations, total hospitalization durations, and overall mortality within 28 days of the included patients were retrospectively evaluated and recorded. Patients’ scoring results and demographic characteristics are provided in Appendix A (refer to the Appendix A).

### Statistical Analysis

Statistical analysis was performed using IBM SPSS Statistics Version 25, after initial data collection was completed in Microsoft Excel. All parameters were investigated using descriptive analysis, in which means and standard deviations (SD) were reported for parametric values, while medians, 25th, and 75th percentile values were used for non-parametric values. To assess whether a result is distributed parametrically or not, histogram charts were primarily used along with Kolmograv–Smirknov analysis for confirmation when required. For values deemed parametric, comparisons between two groups were made using an independent samples *t*-test after evaluating linearity with Shapiro–Wilk’s test and confirming equality of variance with Levene’s test. For correlation analysis, Pearson correlation was utilized. Receiver Operating Characteristic (ROC) Curve analyses were presented with charts and p-scores, with *p*-values compared to a 50% area under the curve (AUC) assumption.

## 3. Results

A total of 740 patients were evaluated for the study. Those diagnosed with sepsis and septic shock (n = 165, 22%) were accepted as the study group. The majority of patients (n = 110, 66.7%) were male, with an average age of 70.3 (±15.8) years. The patients had an average Charlson Comorbidity Index (CCI) score of 6.88 (±2.66). A median of 12 (6–22) days for hospitalization and 3 (1–6) days of ICU admission was reported, with less than half (n = 71, 43%) of patients requiring inotropic support and a median of 1 day (0–5) on invasive mechanical ventilation history. White blood cell levels were elevated, with a mean of 13.0 109/L (7.9–19.7), and neutrophils were observed at 83.8% (±16.2). Correlated with sepsis diagnosis, procalcitonin and C-reactive protein levels were increased at 4.9 ng/mL (2.9–13.9) and 15.6 mg/dl (8.2–23.0), respectively. All mortality scoring systems were found to be increased, with APACHE II, SOFA, SAPS II, and OASIS being reported at 26.17 (±7.96), 8.0 (±3.0), 56.73 (±14.53), and 34.38 (±10.0), respectively. The 28-day mortality rate was 63.6% (105) (Table 1).

Comparing parameters between the exitus group and the survival one showed that there was a significant difference in gender, age, CCI score, hospital admission days, ICU admission days, days on mechanical ventilation and inotropic support, with female gender and longer hospital admission duration being observed more commonly in the survival group and the rest of the parameters being more common in the exitus group. There was also significance in all mortality scoring parameters, with all parameters being higher in the exit group (Table 2).

All parameters having a statistical difference regarding mortality were then evaluated by correlation analysis. All four scoring systems, APACHE II, SOFA, SAPS II, and OASIS, positively correlated with mortality and CCI score. Excluding OASIS, a negative correlation was observed between hospitalization duration and the scoring system. Only OASIS correlated with ICU admission duration and mechanical ventilation duration, which was found to be positive in both cases. Gender only correlated with SOFA scoring, while age positively correlated with SAPS II and OASIS (Table 3).

Receiver operating characteristic (ROC) curves were created to evaluate the role of mortality scoring systems, in which all scoring system models had statistical relevance (p score of 0.001 for all analyses). APACHE II had the lowest AUC at 0.803, followed by SOFA with 0.873, OASIS at 0.879, and the highest being SAPS at 0.903. All models had relatively small standard errors, ranging between 0.027 and 0.033 (Table 4 and Table 5) (Figure 1).

Regarding pairwise comparison, differences in AUC between scoring systems were compared by Delong et al.’s calculation of the Standard Error of AUC and of the difference between two AUCs. SOFA, SAPS II, and OASIS had a higher AUC compared to APACHE II (*p* value of 0.024, 0.011, and 0.049, respectively). No other statistically significant difference was observed between other scoring system comparisons, indicating comparable discriminative performance.

## 4. Discussion

The study investigated and compared the scoring systems of SOFA, SAPS II, APACHE II, and OASIS in their respective roles in predicting mortality and prognosis in patients admitted to the ICU with a diagnosis of sepsis or septic shock. Comparisons of these scores were deemed informative, especially regarding their possible role in a setting where time of essence, such as sepsis [4,5,6].

Sepsis is defined as an irregulated response of the host to an infection, resulting in organ dysfunction. Septic shock, as stated in the inclusion criteria, could be described as a vasopressor requirement despite adequate fluid resuscitation to ensure a MAP at or above 65 mmHg and/or a serum lactate level at or above 2 mmol/L (>18 mg/dL) [20].

As expected, we found significance in all mortality scoring parameters in our study. OASIS correlated with the length of stay in the intensive care unit and duration of mechanical ventilation. SAPS II and OASIS were found to have higher correlations with mortality than other scoring systems.

The Acute Physiology and Chronic Health Evaluation (APACHE) II, Simplified Acute Physiology Score (SAPS) II, Sequential Organ Failure Assessment (SOFA) score, and Oxford Acute Severity of Illness Score (OASIS) are widely used as severity-of-illness scores in intensive care units (ICUs) to assess patient severity, predict outcomes, and guide clinical management.

The APACHE II score, developed in 1985, incorporates 12 physiological variables and a chronic health evaluation to predict ICU mortality and disease severity [21]. Its complexity, however, has led to the development of the SAPS II, which simplifies the assessment by reducing the number of variables while maintaining comparable accuracy in predicting patient outcomes [22]. The SOFA score, introduced in the 1990s, assesses organ dysfunction across six organ systems, providing a dynamic measure that can track disease progression [21]. It is particularly useful for assessing sepsis-related organ failure and is a key component in the Sepsis-3 criteria [20]. The OASIS does not require laboratory tests or imaging examinations, making it a practical option for rapid assessment. It evaluates parameters such as age, Glasgow Coma Scale score, mean arterial pressure, and reason for ICU admission to predict patient outcomes. The OASIS also provides a measure of acute severity and has been found to be effective in predicting mortality across diverse ICU populations. While all these scores aim to predict patient outcomes and guide care, their clinical applicability can vary. APACHE II and SAPS II are more complex and better suited for prognosis in a broad ICU context, whereas SOFA and OASIS are more focused on tracking organ dysfunction and acute severity. Each scoring system has its strengths and weaknesses, and their utility depends on specific clinical goals, including predicting mortality, guiding treatment, and resource allocation [16].

APACHE II and SOFA scores are easily calculated from data routinely available during the first 24 h of admission and have been extensively studied in general ICU populations [21].

SOFA, SAPS II, and APACHE II are solid and reliable mortality scoring systems. In our study, an APACHE II score of 26 (estimated mortality rate 55%), SOFA score of 8 (estimated mortality rate 33.3%), and SAPS II score of 56.7 (estimated mortality rate 61.9%) were observed, with the actual mortality being reported at 63.6%. Similarly, in the study conducted by Ari et al., high APACHE II and SOFA scores were associated with mortality [23]. In our study, the most significant difference between the actual and estimated mortality was seen in SOFA scoring. This difference was confirmed in the analysis performed between survivors, as all four systems had statistically significant differences between them. All four scoring systems had lower mortality correlated with a lower overall score. The presence of higher mortality in patients with inotropic support requirements and/or mechanical ventilation requirements also confirmed the four scores’ validity and led to the assumption that the parameters included in the study were robust.

The study of Chen et al. evaluated the performance of OASIS in assessing mortality in septic patients, comparing it with the Sepsis-related Organ Failure Assessment (SOFA) score, and OASIS was observed to be correlated with clinical outcomes in patients with sepsis; however, SAPS II was reported to be superior in predicting mortality [15]. Jia et al., in a similar study, showed that age was the most prominent factor in terms of all causes of mortality [24]. Age was a part of scoring systems, excluding SOFA, in our study and was lower in the survival group.

In many studies, SOFA was considered a valuable factor in estimating the prognosis of patients with sepsis or ICU care [25,26,27]. In the study conducted by Tüten et al., mortality was significantly increased in patients with high APACHE-II and SOFA scores [28]. However, in Granholm et al.’s study, SAPS II was stated to be superior to SOFA regarding in-hospital mortality and 90-day all-cause mortality [27]. SOFA was also reported to be inferior to OASIS, SAPS II, and APACHE II in terms of mortality estimation in the multi-center study of Wang et al. [29]. The same study suggested the usage of OASIS due to the ease of calculation [29].

Another study highlighted SAPS II’s ability to estimate mortality without the requirement of the initial admission diagnosis, providing another difference between scoring systems [30].

Our study revealed results favoring those discussed, as SAPS II followed by OASIS was more prominent in mortality prediction, with SOFA having the highest difference between the four groups in estimated and real mortality results.

OASIS was created with a reduction in parameters required for calculation while keeping the prediction reliability in mind, by utilizing APACHE IV as the basis. Chen et al. stated that OASIS was simpler to calculate, with fewer laboratory parameters being required in estimation [15]. Zhu et al. reported that OASIS had a high association between 28-day mortality in patients with sepsis [31]. Huang et al. showed that OASIS was correlated with the prognosis of ICU patients with respiratory failure and also had prognostic value for mortality [32]. Our study confirmed this assumption, as OASIS was practical and accessible enough. However, a note of interest and perhaps limitation was that OASIS was the sole scoring system that included ICU admission duration and mechanical ventilation requirement history. These in-built requirements could have led to the assumption that OASIS is only valid for evaluating sepsis patients only in the ICU setting. In our study, the ROC analysis of all four scoring systems was statically significant and similar in shape, with SAPS II having the highest AUC, followed by OASIS.

This supported the correlation analysis, as while all scoring systems showed higher mortality as the points assigned to them increased, the highest correlation between mortality and scores was observed in SAPS II, followed by OASIS. In another study we conducted, it was found that high CCI, APACHE II, and SOFA scores significantly increase mortality, and disease severity, age, and infection in intensive care are important factors affecting mortality [33]. Furthermore, the study conducted by Kao et al. also supports this [34]. Similarly, CCI scores and scoring systems had a positive correlation, which was an expected observation, as all scoring systems either had comorbidity evaluation or comorbidity-related laboratory results.

In their 2020 study on intensive care severity scoring systems, Hu C. et al. demonstrated that SOFA, SAPS-II, OASIS, and LODS scores are significantly associated with ICU mortality among patients with sepsis. Notably, SAPS-II and OASIS exhibited superior prognostic performance compared to SOFA and LODS [35]. Similarly, in our study, SAPS-Ⅱ and OASIS have better association with mortality. 

In a study conducted in 2022, covering patients in medical ICU, cardiovascular surgery ICU, and neurological ICU, they found SAPS II and OASIS to be successful in predicting 28-day mortality, similar to our study [36].

Excluding OASIS, all scoring systems negatively correlated with overall hospitalization duration. This could be attributed to the fact that, as the scores increased, the overall mortality of the patients increased, thus reducing the days spent in the hospital or an urgent admission to the ICU due to the severity of the patients. As stated above, OASIS was also the only system not related to the former hospitalization duration and limited to the ICU, and thus, this could have contributed to the lack of negative correlation observed between OASIS and total hospitalization duration.

Wang L et al. found that, in their study on three scoring systems, SAPS II has a good prognostic value in predicting the in-hospital mortality of intensive care patients with STEMI. In terms of predictive value, SAPS II was higher than the probability. However, this group of patients may not have a pre-length of ICU stay because they were admitted to elective surgery and emergency procedures. Therefore, it does not show similarity with the sepsis patients in our study [13].

In our study gender only correlated with SOFA scoring, while age positively correlated with SAPS II and OASIS.

The study’s limitations could be summarized as the limited patient count and its retrospective nature. We believe this was at least partially compensated by the routine scoring methods used in intensive care, including SOFA and APACHE II. These two scoring systems were routinely performed on every patient, while OASIS and SAPS II were calculated from the patients’ files and computer records. As such, the retrospective aspect of the study was primarily in design rather than data investigation, with nearly all data already available from patient follow-up. While statistically adequate in terms of the parametric count, a larger patient population could have allowed for subgroup analysis according to parameters not available in these scoring systems, such as rare comorbidities, which is another limitation of our study.

## 5. Conclusions

Compared to commonly used scoring systems, OASIS is a practical tool and serves as a robust scoring system for assessing mortality in ICU patients diagnosed with sepsis. Excluding SAPS II, OASIS showed a higher correlation with mortality compared to other scoring systems, performed similarly to other scoring systems in terms of mortality estimation, and its components allowed correlation with ICU admission and mechanical ventilation, which was not observed in other scoring systems.

## Figures and Tables

**Figure 1 diagnostics-15-01660-f001:**
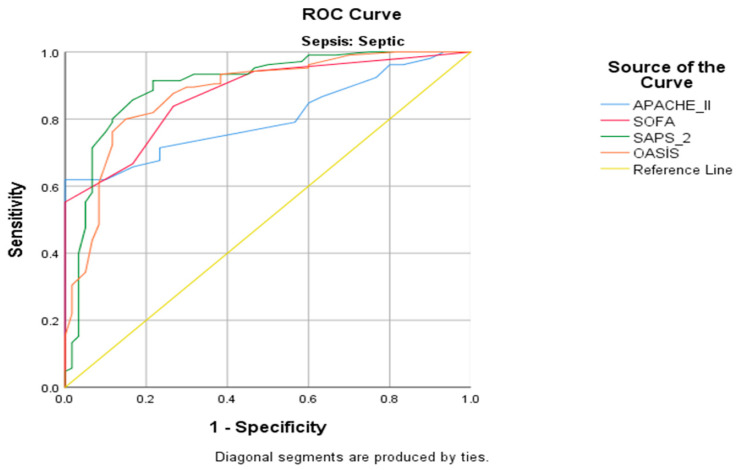
ROC Analysis of Scoring Systems.

**Table 1 diagnostics-15-01660-t001:** Demographic Characteristics and Laboratory Parameters of Patients.

Parameters (n, %)		Patients (n = 165)
Gender	Male (%)	110 (66.7)
	Female (%)	55 (33.3)
Age (years, SD)		70.38 (±15.85)
Charlson Comorbidity Index (SD)		6.88(±2.66)
Hospital Admission Days		12 (6–22)
ICU Admission Days		3 (1–6)
Days on Mechanical Ventilation		1 (0–5)
Inotropic Support Requirement (%)		71 (43)
White Blood Cell (10^9^/L)		13.0 (7.9–19.7)
Neutrophile (10^9^/L)		10.5 (5.6–15.8)
Neutrophile (%, SD)		83.8 (±16.25)
Procalcitonin (ng/mL)		4.9 (2.9–13.9)
C-Reactive Protein (mg/L)		15.6 (8.2–23.9)
Mortality Scoring and Overall Mortality		
APACHE II		26.17 (±7.96)
SOFA		8.0 (±3.0)
SAPS II		56.73 (±14.53)
OASIS		34.38 (±10.0)
28-day Mortality (%)		105 (63.6)

SD: Standard Deviation, ICU: Intensive Care Unit, APACHE II: Acute Physiology and Chronic Health Evaluation II, SOFA: Sequential Organ Failure Assessment, SAPS II: Simplified Acute Physiology Score II, OASIS: Oxford Acute Severity of Illness Score. The 25th–75th refers to the 25th and 75th percentile; the values given here are median values.

**Table 2 diagnostics-15-01660-t002:** Comparison between parameters regarding 28-day mortality.

Independent Samples *t*-Test	t	dF	*p*	95% CI of the Difference	
				Lower	Higher
Gender	2.015	112	0.046 *	0.003	0.312
Age (years)	−4.590	86	0.001 *	−17.675	−6.992
Charlson Comorbidity Index	−9.327	155	0.001 *	−3.716	−2.417
Hospital Admission Days	3.950	94	0.001 *	4.436	13.402
ICU Admission Days	−2.270	160	0.025 *	−4.488	−0.312
Days on Mechanical Ventilation	−3.495	161	0.001 *	−6.104	−1.696
Inotropic Support Requirement	−8.428	161	0.001	−0.641	−0.397
White Blood Cell (10^9^/L)	−0.337	156	0.737	−4.338	3.075
Neutrophile (10^9^/L)	−1.221	148	0.224	−5.276	1.246
Neutrophile (%)	−1.762	163	0.080	−9.768	0.556
Procalcitonin	−1.732	163	0.085	−12.387	0.809
C-Reactive Protein	0.475	163	0.636	−4.026	6.576
APACHE II	−8.777	162	0.001 *	−10.401	−6.580
SOFA	−10.512	155	0.001 *	−4.231	−2.893
SAPS II	−10.637	163	0.001 *	−22.854	−15.698
OASIS	−10.553	163	0.001 *	−15.780	−10.806

CI: Confidence interval, dF: Degrees of Freedom, ICU: Intensive Care Unit, APACHE II: Acute Physiology and Chronic Health Evaluation II, SOFA: Sequential Organ Failure Assessment, SAPS II: Simplified Acute Physiology Score II, OASIS: Oxford Acute Severity of Illness Score, * significant *p* values.

**Table 3 diagnostics-15-01660-t003:** Correlation Between Scoring Systems, Mortality, and Prognostic Factors.

Parameters	Pearson Correlation	APACHE II	SOFA	SAPS II	OASIS
Gender	Correlation	−0.106	−0.178 ^a^	−0.149	−0.018
	*p*-value	0.176	0.022 *	0.057	0.823
Age	Correlation	0.102	0.145	0.430 ^b^	0.360 ^b^
	*p*-value	0.194	0.063	0.001 *	0.001 *
Charlson Comorbidity Index	Correlation	0.340 ^b^	0.370 ^b^	0.613 ^b^	0.470 ^b^
	*p*-value	0.001 *	0.001 *	0.001 *	0.001 *
Hospital Admission Days	Correlation	−0.169 ^a^	−0.245 ^b^	−0.226 ^b^	−0.138
	*p*-value	0.030 *	0.002 *	0.003 *	0.077
ICU Admission Days	Correlation	−0.071	−0.078	0.085	0.271 ^b^
	*p*-value	0.365	0.320	0.280	0.001 *
Days on Mechanical Ventilation	Correlation	0.020	0.059	0.139	0.339 ^b^
	*p*-value	0.796	0.454	0.076	0.001 *
28-day Mortality	Correlation	0.515 ^b^	0.564 ^b^	0.640 ^b^	0.637 ^b^
	*p*-value	0.001 *	0.001 *	0.001 *	0.001 *

ICU: Intensive Care Unit, APACHE II: Acute Physiology and Chronic Health Evaluation II, SOFA: Sequential Organ Failure Assessment, SAPS II: Simplified Acute Physiology Score II, OASIS: Oxford Acute Severity of Illness Score. ^a^ There is a weak correlation between the variables, ^b^ There is a moderate to strong correlation between the variables, * significant *p* values.

**Table 4 diagnostics-15-01660-t004:** Receiver Operating Characteristic Curve Tables of Scoring Systems and 28-day Mortality.

	Area Under Curve	Standard Error	*p*	%95 Confidence Interval	
				Lower Bound	Upper Bound
APACHE II	0.803	0.033	0.001 *	0.738	0.868
SOFA	0.873	0.027	0.001 *	0.821	0.925
SAPS II	0.902	0.027	0.001 *	0.849	0.955
OASIS	0.879	0.028	0.001*	0.823	0.935

AUC: Area under curve, APACHE II: Acute Physiology and Chronic Health Evaluation II, SOFA: Sequential Organ Failure Assessment, SAPS II: Simplified Acute Physiology Score II, OASIS: Oxford Acute Severity of Illness Score. *p* value is reported to a test of AUC over 0.5, * significant values.

**Table 5 diagnostics-15-01660-t005:** Paired sample area difference under ROC curves.

Test Pairs	AUC Difference	95% Confidence Interval		Standard Score	Standard Error Difference ^1^	*p*-Value
		Lower	Upper			
APACHE II—SOFA	−0.070	−0.131	−0.009	−2.251	0.242	0.024 *
APACHE II—SAPS II	−0.099	−0.176	−0.023	−2.557	0.245	0.011 *
APACHE II—OASIS	−0.076	−0.152	0.000	−1.957	0.247	0.049 *
SOFA—SAPS II	−0.030	−0.092	0.033	−0.921	0.230	0.357
SOFA—OASIS	−0.006	−0.066	0.054	−0.200	0.233	0.841
SAPS II—OASIS	0.023	−0.027	0.074	0.908	0.234	0.364

AUC: Area under the curve, APACHE II: Acute Physiology and Chronic Health Evaluation II, SOFA: Sequential Organ Failure Assessment, SAPS II: Simplified Acute Physiology Score II, OASIS: Oxford Acute Severity of Illness Score. ^1^ Given for evaluation under nonparametric assumption, * significant *p* values.

## Data Availability

All relevant data are within the paper and its Appendix A.

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
