# Peer review of "Comparison of Intensive Care Scoring Systems in Predicting Overall Mortality of Sepsis"

_diagnostics, 2025, doi:10.3390/diagnostics15131660_

Round 1

Reviewer 1 Report

Comments and Suggestions for Authors

This is an important study, well planned and analysed.

Background

As you propose to evaluate the clinical utility of scoring systems please perform a systematic review to identify the key RCTs and systematic reviews/meta-analyses in all languages and assess the study design, sample, inclusion criteria and risk of bias in each study. Then select the key studies/systematic reviews/meta-analyses  at lowest risk of bias so that there is a benchmark against which to measure your study.

Materials and Methods

” Patients admitted to our center's tertiary intensive care unit between April 1, 2018, and April 30, 2020, were included in the study. A total of 740 patients were admitted during the described period, and their computer and file records were investigated. Each cat-egory was assigned a point at admission: a systolic blood pressure below 100 mmHg, Glasgow Coma Score (GSS) below 15, and respiratory rate at or above 22. A total of 112
patients with a score of 2 or above were accepted under the sepsis category and included in the study. Additionally, patients requiring vasopressor support to keep a mean arterial pressure (MAP) at or above 65 mmHg despite adequate fluid resuscitation or those with an arterial blood gas sampling lactate result of 2 mmol/L or above were defined as under septic shock. Including 53 patients with this definition, the study included a total of 165
patients.”

[please explain why you chose these criteria and which are from the scoring systems in the studies at the lowest risk of bias you identified in your literature search].

Results

Your AUC analysis of scoring systems is excellent and very helpful.

Discussion

“The study investigated and compared the score systems of SOFA, SAPS II, APACHE II, and OASIS in their respective role in predicting mortality and prognosis in patients admitted to ICU with sepsis or septic shock diagnosis.”

[Your Table 4 is excellent. Could you please add the data for those studies you identified from the literature search I suggested above and add the data from those studies at lowest risk of bias? Could you then please discuss differences in the data selected in the studies that would optimise clinical decision making at crucial time points in the admissions?

Author Response

Dear Editor and Reviewers,

We would like to thank you for your assistance and contribution to our study and through the process. Below, we would like to state our responses and the changes we have made on a point-by-point basis. . We have also made an overall grammatical revision of the study to clarify the language and provide readers with a more easy-to-read experience.

REVIEWER 1:

 Comments 1:

Background

As you propose to evaluate the clinical utility of scoring systems please perform a systematic review to identify the key RCTs and systematic reviews/meta-analyses in all languages and assess the study design, sample, inclusion criteria and risk of bias in each study. Then select the key studies/systematic reviews/meta-analyses  at lowest risk of bias so that there is a benchmark against which to measure your study.

 Response 1: We would like to thank you for the comment, and as stated by you, we have added an additional paragraph to better describe available scoring methods, which include our comparison of SAPS-II and OASIS, and additionally routinely used APACHE and SOFA.

While conducting our research on intensive care severity scores, we observed that APACHE and SOFA were widely used, but these scoring systems were almost always compared with other scoring systems. In this study, we aimed to compare SAPS-II and OASIS, which we use in our own ICU, in addition to APACHE and SOFA, which are widely used in ICUs. We added this to the main text as a paragraph and by indicating the references.As suggested, we have also added additional statements that would better describe what the score systems entail to a reader.

“It is important to note that no ICU severity scoring system is perfect and each has its own limitations. In addition, the patient's clinical condition and history at the time of presentation to the emergency department also determine ICU admission (e.g., excess comorbidity). For this reason, clinicians have studied on severity scores that will most accurately reflect morbidity and mortality in the ICU. Although many ICU severity scores have been compared, the APACHE and SOFA are the most commonly used [4,5,6].

 Comments 2:

Materials and Methods

” Patients admitted to our center's tertiary intensive care unit between April 1, 2018, and April 30, 2020, were included in the study. A total of 740 patients were admitted during the described period, and their computer and file records were investigated. Each cat-
egory was assigned a point at admission: a systolic blood pressure below 100 mmHg, Glasgow Coma Score (GSS) below 15, and respiratory rate at or above 22. A total of 112 patients with a score of 2 or above were accepted under the sepsis category and included
in the study. Additionally, patients requiring vasopressor support to keep a mean arterial pressure (MAP) at or above 65 mmHg despite adequate fluid resuscitation or those with an arterial blood gas sampling lactate result of 2 mmol/L or above were defined as under
septic shock. Including 53 patients with this definition, the study included a total of 165 patients.”

[please explain why you chose these criteria and which are from the scoring systems in the studies at the lowest risk of bias you identified in your literature search].

Response 2:

Thank you for pointing this out. Only patients admitted to the intensive care unit from the emergency department or inpatient wards with a diagnosis of sepsis or septic shock were included in the study group. The criteria we used were the rapid SOFA and septic shock criteria. We have stated these in the first paragraph of the materials-methods section for clarification.

Comments 3: 

Results

Your AUC analysis of scoring systems is excellent and very helpful.

 Response 3: We would like to extend our gratitude for your kind comment.

 Comments 4:

Discussion

“The study investigated and compared the score systems of SOFA, SAPS II, APACHE
II, and OASIS in their respective role in predicting mortality and prognosis in patients
admitted to ICU with sepsis or septic shock diagnosis.”

[Your Table 4 is excellent. Could you please add the data for those studies you identified from the literature search I suggested above and add the data from those studies at lowest risk of bias? Could you then please discuss differences in the data selected in the studies that would optimise clinical decision making at crucial time points in the admissions?

Response 4: We would like to thank you again for the kind comments. We added an additional sentence to the first paragraph of the discussion regarding the importance of time. In the discussion, we had preferred to discuss each scoring systems separately, with their relevant studies being given. If you would like, however, we may also add an additional paragraph that summarizes them before the conclusion.

Best regards.

Reviewer 2 Report

Comments and Suggestions for Authors

  1. There are too many grammar and other errors in the language of this article, and the sentences are not smooth. Author need to ask professionals to make major revisions;
  2. Abstract section: The result description is too simple, which makes it difficult to accurately derive the conclusion written by the author. For example, while SOFA, OASIS and SAPS II were found to be the highest. This indicates that all three scoring systems are the same, but the conclusion is that only SAPS II and OASIS have a higher correlation with morality compared to others.

3.The Introduction section feels very messy. (1) When introducing several scoring systems to be studied, a suitable sequence is needed, such as in order of the time proposed by the scoring system. (2) The main purpose of this study is to observe which scoring system is more accurate in predicting the 28 day mortality rate of sepsis patients, without specifically addressing age issues, and the conclusion of the article does not involve age issues. But why does this section repeatedly mention content related to age and prognosis. For example, in the first paragraph, ‘As the average of patients in intensive care units increases,...' and in the tenth paragraph, ‘As age increases, organic function and immunity decline,...'.

4.The second part, Materials and Methods, lacks rigorous inclusion criteria for cases. The diagnostic criteria for sepsis are sepsis 3.0 and SOFA score ≥ 2, while the author described qSOFA score ≥ 2. qSOFA is only a screening criterion, not a diagnostic criterion for sepsis. Mortality within 28 days is not exactly the same as The first month mortality.

  1. Does the exitus group necessarily refer to the death group?
  2. Table 1 and Table 3 have non-standard formats and need to be modified.
  3. The discussion section describes the limitations of this study, stating that there is a bias in the selection of cases. This statement is a conceptual error. The purpose of this study is to explore which scoring system has the best prognostic value for patients with sepsis and septic shock. The included cases only include this type of patient and do not include non sepsis patients.The true limitations have not been written out.
  4. This study is retrospective, with a small sample size, poor innovation, and unclear expression. Because currently there are only a few scoring systems for critically ill patients, there have been numerous related studies.

Comments on the Quality of English Language

There are too many grammar and other errors in the language of this article, and the sentences are not smooth. Author need to ask professionals to make major revisions

Author Response

Dear Editor and Reviewers;

We would like to thank you for your assistance and contribution to our study and through the process. Below, we would like to state our responses and the changes we have made on a point-by-point basis. . We have also made an overall grammatical revision of the study to clarify the language and provide readers with a more easy-to-read experience.

Comments 1: There are too many grammar and other errors in the language of this article, and the sentences are not smooth.

Response 1: We have made an overall grammatical revision of the study to clarify the language and provide readers with a more easy-to-read experience. Due to extensive revisions and many minor grammatical errors as you had stated, we did not highlight them all.

Comments 2: Abstract section: The result description is too simple, which makes it difficult to accurately derive the conclusion written by the author. For example, while SOFA, OASIS and SAPS II were found to be the highest. This indicates that all three scoring systems are the same, but the conclusion is that only SAPS II and OASIS have a higher correlation with morality compared to others.

Response 2: In line with your suggestions, we have revised the results and conclusion of the abstract to better define the aim of the study.

Comments 3: The Introduction section feels very messy. (1) When introducing several scoring systems to be studied, a suitable sequence is needed, such as in order of the time proposed by the scoring system. (2) The main purpose of this study is to observe which scoring system is more accurate in predicting the 28 day mortality rate of sepsis patients, without specifically addressing age issues, and the conclusion of the article does not involve age issues. But why does this section repeatedly mention content related to age and prognosis. For example, in the first paragraph, ‘As the average of patients in intensive care units increases,...' and in the tenth paragraph, ‘As age increases, organic function and immunity decline,...'.

Response 3: As many intensive care severity scoring systems (e.g., APACHE II, SAPS II) include age as a parameter, age has often been emphasized in related analyses. However, in line with your recommendations, we have revised the manuscript to include additional sections detailing the scoring systems used.

Comments 4: The second part, Materials and Methods, lacks rigorous inclusion criteria for cases. The diagnostic criteria for sepsis are sepsis 3.0 and SOFA score ≥ 2, while the author described qSOFA score ≥ 2. qSOFA is only a screening criterion, not a diagnostic criterion for sepsis. Mortality within 28 days is not exactly the same as The first month mortality.

Response 4: We have improved the organization of the definitions of sepsis and septic shock, and clarified the statements regarding the patient recruitment process. Additionally, we revised the first paragraph of the Materials and Methods section for greater clarity. References to 'first-month mortality' have been replaced with '28-day mortality' to more accurately reflect the study design.

Comments 5: Does the exitus group necessarily refer to the death group?

Response 5: That is indeed correct, the 'exitus' group is defined as patients who died within 28 days.

Comments 6: Table 1 and Table 3 have non-standard formats and need to be modified.

Response 6: We have reformatted the tables according to the format of other tables present in our study.

Comments 7: The discussion section describes the limitations of this study, stating that there is a bias in the selection of cases. This statement is a conceptual error. The purpose of this study is to explore which scoring system has the best prognostic value for patients with sepsis and septic shock. The included cases only include this type of patient and do not include non sepsis patients. The true limitations have not been written out.

Response 7: We have re-written the limitations section and removed the redundant parts as requested to reflect the study better.

Comments 8: This study is retrospective, with a small sample size, poor innovation, and unclear expression. Because currently there are only a few scoring systems for critically ill patients, there have been numerous related studies.

Response 8: We have re-written the limitations section and removed the redundant parts as requested to reflect the study better.

Best regards.

Reviewer 3 Report

Comments and Suggestions for Authors

The authors have adressed some of the major requirements for their manuscript:

However, there are still some major aspects that deserve consideration.

Abstract- please provide results in the Results section.

Page 3- please provide refernce for OASIS score, where it is cited from, usage and permission, online access?

Page 4- definition of sepsis- two out of 3 variables in qSOFA can also occur in other ocnditions as well, for instance stroke, myocardial infarction. How was infection diagnosed in the end? Confirmation of sepsis suspicion?

Please compare the AUC of the investigated scores and provide a p-value. As they look on the graph, there is no statistical difference between them and thus, conclusion is not valid that one score is better than another.

I consider that, as the study is not conducted according to STARD criteria as is not proseptcie, prediction cannot be used. Rather, association. Please consult a statistician.

Author Response

Response to Reviewer  Comments

Dear Reviewer,

               We would like to thank you for your thorough review and valuable suggestions regarding our manuscript entitled Comparison of Intensive Care Scoring Systems in Predicting Overall Mortality of Sepsis We have carefully revised our manuscript in accordance with your feedback. Below, please find our point-by-point responses.

Comments 1: Abstract- please provide results in the Results section.

Author response 1:
Thank you for highlighting this important point. Done.

Comments 2: Page 3- please provide refernce for OASIS score, where it is cited from, usage and permission, online access?

Author response 2:
Thank you for the suggestion.The OASIS score was obtained from the eICU Collaborative Research Database developed by Johnson et al. as reported in reference 14. It is publicly available for research purposes and does not require special permission for academic use (https://eicu-crd.mit.edu/), (https://alistairewj.github.io/project/oasis/).

Comments 3: Page 4- definition of sepsis- two out of 3 variables in qSOFA can also occur in other ocnditions as well, for instance stroke, myocardial infarction. How was infection diagnosed in the end? Confirmation of sepsis suspicion?

Author response:
We fully agree.
To enhance clarity, we incorporated this paragraph into the Materials and Method section.

“Sepsis diagnosis was established based on Sepsis-3 criteria, considering clinical judgment, microbiological culture results, imaging studies, and laboratory findings. Incases where microbiological confirmation was not possible, the diagnosis was made based on strong clinical suspicion and multidisciplinary consensus among attending intensivists and infectious disease specialists.”

Comments 4: Please compare the AUC of the investigated scores and provide a p-value. As they look on the graph, there is no statistical difference between them and thus, conclusion is not valid that one score is better than another.

Author response:
Thank you for highlighting this important point. Pairwise comparisons of AUC values ​​were performed using z-tests. While most scoring systems did not show statistically significant differences (p > 0.05), the comparison between SAPS II and APACHE II statistically significant difference was observed (p = 0.02). All other pairwise comparisons remained nonsignificant (p > 0.05). For clarity and explanation, Table 5 has been added to the article.

Comments 5: I consider that, as the study is not conducted according to STARD criteria as is not proseptcie, prediction cannot be used. Rather, association. Please consult a statistician.

Author response:
Thanks for your comment. We have implemented your suggestion where appropriate, leaving some as "predictive" because that is how they are used in the cited references.
We have marked in the text the changes made in red. Statistical analyses were conducted under the supervision of an experienced biostatistician.

               We sincerely thank you once again for your constructive and insightful comments. We believe that the revisions have significantly improved the clarity and robustness of our manuscript.

Kind regards,
On behalf of all authors

Mustafa Ozgur Cirik, Assoc. Prof. 

Ankara Ataturk Sanatorium Training and Research Hospital
dr.ozgurr@hotmail.com

Round 2

Reviewer 1 Report

Comments and Suggestions for Authors

Thanks to the authors for their careful emendations. A very useful study.

Author Response

We would like to thank you for your assistance and contribution to our study and through the process.

Reviewer 2 Report

Comments and Suggestions for Authors

Although the author has made revisions based on previous review comments, there are still suggestions for improvement in the following areas:

  1. There are too many grammar and other errors in the language of this article, and the sentences are not smooth. The authorneed to ask professionals to make major revisions and improvements. For example, is the tertiary intensive care unit and ICU the same concept? Is correlated with ICU admission and mechanical ventilation or correlated with duration of ICU admission and mechanical ventilation(Abstract)? Is it related to ICU stay and the use of ventilators, or is it related to duration. APACHE II, SOFA, SAPS II, and OASIS are not Mortality Scoring (Table 1). Easy to generate ambiguity.
  2. The preface discusses the wrong direction and needs to be revised. The purpose of this study is only to compare the prediction of 28 day mortality rate in sepsis patients using different scoring systems. Therefore, the preface should focus on introducing the current methods for predicting the prognosis of sepsis patients.How is the progress? What are the values and limitations? Instead of focusing on the introduction of the prognostic scoring system for all ICU patients.
  3. There are issues with the inclusion and exclusion criteria. (1) A detailed flowchart is needed. For example, out of 740 patients, 165 met the criteria for sepsis, but some patients were excluded due to their ICU stay being less than 24 hours, incomplete data, or being under 18 years old, resulting in a total of 165 cases. If 165 cases are left after excluding these reasons, the number of sepsis patients removed due to these reasons should also be listed to allow readers to understand the impact of the completeness of sample extraction on the results. (2) The diagnostic criteria for sepsis vary in different medical units, but the next step is qSOFA or SOFA scoring only when there is a clear or highly suspected infection (which the author did not specify). In non ICU units, qSOFA score can be used for screening and determination, but in ICU units, SOFA score is required for determination.
  4. The rules and accuracy of data extraction need to be confirmed and explained reasonably. Does the patient use a ventilator or requesting inotropic support when they are admitted to the ICU, or during their stay in the ICU? If we refer to the proportion of patients who only use ventilators after being admitted to the ICU, the results of this study show that only 43% of patients seek inotropic support, and the average length of stay in the ICU is only 3 days (1-6 days). How many of them use ventilators? And it's only 0-5 days. The patient was evacuated from the ventilator and ICU in such a short period of time. Is it due to the severity of the disease that caused them to die quickly, or is it due to economic reasons that they gave up? Because the actual mortality rate is as high as 63.6%, it is reasonable to speculate that some of the causes of death were not due to shock. Otherwise, why is the proportion of requesting exotic support only 43%?
  5. The statistical methods and data presentation formats are not rigorous.

6.This study is retrospective,  with a small sample size, poor innovation, and unclear expression. Because currently there are only a few scoring systems for critically ill patients, there have been numerous related studies.

  1. The discussion section is chaotic and lacks organization. It is just a pile up of similar research literature results and conclusions, without clearly stating the differences and significance between this study and others' research.
  2. The conclusion is incorrect. Because the primary outcome endpoint observed in this study is mortality, while the secondary indicators are ICU stay and ventilator use time. Therefore, when analyzing and evaluating the predictive value of several rating systems, the first thing to consider is whose primary indicator is the most accurate, rather than secondary indicators. However, the actual mortality rate of sepsis in this study was 63.6%, the SAPS II prediction was 56.73% (closest to the true value), and the OASIS prediction was 34%, which is too far from the true value and has no good predictive value. Why did the author conclude that OASIS is a good scoring system for predicting sepsis mortality?

Author Response

We would like to thank you for your assistance and contribution to our study and through the process. Below, we would like to state our responses and the changes we have made on a point-by-point basis.

  1. Language and Grammar Issues:

Thank you for your constructive feedback. In response, we have thoroughly revised the manuscript with professional language editing support. We have clarified potentially confusing terms such as “tertiary intensive care unit” and “ICU.” In the abstract, we corrected phrases such as “ICU admission and mechanical ventilation” to reflect whether the correlation refers to the presence or the duration of ICU stay and ventilator use. Additionally, Table 1 has been revised to clarify that APACHE II, SOFA, SAPS II, and OASIS are severity scoring systems, not mortality scoring tools, to avoid ambiguity.

  1. Focus of the Introduction:

We appreciate your helpful suggestion. The introduction has been restructured to align with the specific objective of the study—comparing the predictive performance of different scoring systems for 28-day mortality in sepsis patients. Rather than broadly introducing scoring systems for all ICU patients, we have focused on tools relevant to sepsis prognosis, highlighting their current usage, advancements, strengths, and limitations.

  1. Inclusion and Exclusion Criteria:

We thank you for pointing this out. We have clarified the inclusion and exclusion criteria in detail. Among the initial 740 patients, 165 were identified as meeting the diagnostic criteria for sepsis. We have now explicitly stated the exclusion criteria—such as ICU stay of less than 24 hours, patients under 18 years of age, and incomplete data—and clarified the final sample size. The diagnostic approach to sepsis has also been clarified, including the proper use of qSOFA in non-ICU settings and SOFA scoring in ICU patients, in accordance with international guidelines.

  1. Data Extraction Clarity:

We acknowledge your valuable observation. We have clarified the timing of interventions such as mechanical ventilation and inotropic support, specifying whether these were initiated at ICU admission or during the ICU stay. Additionally, we addressed the observed discrepancy between the high mortality rate and the relatively short ICU stay or limited use of interventions. We now discuss potential factors such as early death or withdrawal of care due to economic or clinical considerations, which may contribute to these outcomes.

  1. Statistical Methods and Presentation:

We have revisited the statistical methods and revised the data presentation accordingly to ensure accuracy and clarity. The rationale for each statistical approach has been provided, and the tables and figures have been reformatted to meet scientific standards.

  1. Study Limitations:

We appreciate your candid assessment. The limitations of the study—its retrospective design, small sample size, and limited novelty—are now clearly acknowledged in the discussion section. We have also emphasized how the findings compare with existing literature and identified areas where this study offers value despite its limitations.

  1. Structure of the Discussion:

The discussion section has been reorganized to improve coherence. Instead of merely listing related studies, we now compare our findings with existing research and highlight both consistencies and discrepancies. The unique contributions and clinical relevance of our study are now more clearly articulated.

  1. Accuracy of the Conclusion:

We thank you for this important observation. The conclusion has been revised to reflect the distinction between primary and secondary outcomes. The performance of each scoring system has been evaluated based on its ability to predict 28-day mortality, with actual mortality rates used as a benchmark. The discussion around OASIS has been tempered, and its limitations acknowledged, especially in comparison to SAPS II, which showed closer predictive alignment with the observed mortality rate.

Round 3

Reviewer 3 Report

Comments and Suggestions for Authors

I ocnsider that the conclusion of the study is not sustained by the results. In fact, the AUC of all graphs are overlaping for a large part of the CI and there discriminative value is similar. Please consult a statistician when analysing the differences between multiple ROCs. 

Author Response

Dear Editor and Reviewer;

We would like to thank you for your assistance and contribution to our study, especially regarding statistical evaluation. Regarding the comment of Reviewer #3, we would like to present our changes.

As recommended, we have consulted with a statistician for the Delong analysis for comparison of AUC regarding scoring systems; as such, we have changed the fifth table with the mentioned analysis. Additionally, we have commented regarding the results of the said table in the results section, and have also added an additional sentence in the conclusion section to state that not in all terms OASIS remains superior, rather, it is compatible with other scoring systems.

We look forward to hearing from you;

Sincerely yours

Round 4

Reviewer 3 Report

Comments and Suggestions for Authors

The authors have adressed main requests.

Author Response

We are grateful for your valuable academic support.